# Genome-Wide Mining of CULLIN E3 Ubiquitin Ligase Genes from *Uncaria rhynchophylla*

**DOI:** 10.3390/plants13040532

**Published:** 2024-02-15

**Authors:** Yingying Shao, Detian Mu, Yu Zhou, Xinghui Liu, Xueshuang Huang, Iain W. Wilson, Yuxin Qi, Ying Lu, Lina Zhu, Yao Zhang, Deyou Qiu, Qi Tang

**Affiliations:** 1College of Horticulture, National Research Center of Engineering Technology for Utilization of Botanical Functional Ingredients, Hunan Agricultural University, Changsha 410128, China; syy2250519718@126.com (Y.S.); mudetian12580@163.com (D.M.); m18574824089@163.com (Y.Z.); 18627362607@163.com (X.L.); luying960522@163.com (Y.L.); zhulina_2021@163.com (L.Z.); ningzby@163.com (Y.Z.); 2Hunan Provincial Key Laboratory for Synthetic Biology of Traditional Chinese Medicine, Hunan University of Medicine, Huaihua 410208, China; qyx123202110@163.com; 3CSIRO Agriculture and Food, Canberra, ACT 2601, Australia; 4State Key Laboratory of Tree Genetics and Breeding, Research Institute of Forestry, Chinese Academy of Forestry, Beijing 100091, China; qiudy@caf.ac.cn

**Keywords:** *Uncaria rhynchophylla*, CULLIN E3 ubiquitin ligase, expression pattern, ABA stress, TIAs

## Abstract

CULLIN (CUL) protein is a subtype of E3 ubiquitin ligase that is involved in a variety of biological processes and responses to stress in plants. In *Uncaria rhynchophylla*, the *CUL* gene family has not been identified and its role in plant development, stress response and secondary metabolite synthesis has not been studied. In this study, 12 *UrCUL* gene members all contained the typical N-terminal domain and C-terminal domain identified from the *U. rhynchophylla* genome and were classified into four subfamilies based on the phylogenetic relationship with *CULs* in *Arabidopsis thaliana*. They were unevenly distributed on eight chromosomes but had a similar structural composition in the same subfamily, indicating that they were relatively conserved and potentially had similar gene functions. An interspecific and intraspecific collinearity analysis showed that fragment duplication played an important role in the evolution of the *CUL* gene family. The analysis of the *cis*-acting elements suggests that the *UrCULs* may play an important role in various biological processes, including the abscisic acid (ABA) response. To investigate this hypothesis, we treated the roots of *U. rhynchophylla* tissue-cultured seedlings with ABA. The expression pattern analysis showed that all the *UrCUL* genes were widely expressed in roots with various expression patterns. The co-expression association analysis of the *UrCULs* and key enzyme genes in the terpenoid indole alkaloid (TIA) synthesis pathway revealed the complex expression patterns of 12 *UrCUL* genes and some key TIA enzyme genes, especially *UrCUL1*, *UrCUL1-likeA*, *UrCUL2-likeA* and *UrCUL2-likeB,* which might be involved in the biosynthesis of TIAs. The results showed that the *UrCULs* were involved in the response to ABA hormones, providing important information for elucidating the function of UrCULs in *U. rhynchophylla*. The mining of *UrCULs* in the whole genome of *U. rhynchophylla* provided new information for understanding the *CUL* gene and its function in plant secondary metabolites, growth and development.

## 1. Introduction

*Uncaria rhynchophylla*, named “Gou-teng” because of its stem with hooks, is a traditional Chinese medicinal herb used in medicine and clinical practice for many years [1]. It has a good effect on treating neurodegenerative, cardiovascular and cerebrovascular diseases thanks to its active components, terpenoid indole alkaloids (TIAs) [2,3,4]. Although TIAs have important medical value, their biosynthetic pathways are complex and require additional research to elucidate (Figure 1). The biosynthesis of TIAs primarily involves the shikimate and iridoid pathways, which yield two crucial precursors: tryptamine and secologanin [4]. The iridoid pathway involves the catalysis of geranyl diphosphate by geranyl diphosphate diphosphatase (GES) to produce geraniol [5]. Subsequently, geraniol is catalyzed by geraniol 10-hydroxylase (G8H) and 8-hydroxygeraniol dehydrogenase (8HGO) to form 10-Oxogeranial [6,7]. The catalysis of 10-Oxogeranial involves iridodial synthase (IS) and iridodial oxidoreductase (IO), resulting in the formation of 7-Deoxyloganetic acid [8]. Further transformations of 7-Deoxyloganetic acid (7-DLH) include oxidation, reduction, glycosylation and methylation, leading to the formation of Loganic acid [9]. Loganic acid is then converted to Loganin by loganic acid methyltransferase (LAMT), and finally transformed into secologanin by secologanin synthase (SLS) [10,11,12]. Tryptamine is derived from indole through TSA and L-tryptophan decarboxylase (TDC) [13]. The catalytic action of Secologanin and tryptamine results in the formation of strictosidine through strictosidine synthase (STR), which enters the biosynthesis pathway of TIAs and gives rise to components such as rhynchophylline and isorhynchophylline. Rhynchophylline and isorhynchophylline have the highest proportion in the total alkaloids in *U. rhynchophylla*, acting on the cardiovascular and central nervous system, and show diverse pharmacological actions including anti-hypertension, anti-arrhythmic and sedative properties [14,15]. Recent studies have shown that rhynchophylline and isorhynchophylline could be beneficial in the treatment of Alzheimer’s disease, and so could provide great benefit to elderly people worldwide [16]. In addition, the TIA hirsutine has antiviral activity against Dengue virus with the properties of high efficacy and low cytotoxicity [17]. As a result, *U. rhynchophylla* has received international attention. Plant growth and development is an extremely complex process, which is regulated by many biological and abiotic factors, including abscisic acid (ABA). ABA is an important phytohormone which accumulates the content of specialized metabolites. Under the influence of ABA, the content of catharanthine has been shown to reach 1.96 mg/g DW in the transgenic hairy roots and following this, the gene expression was higher in *Catharanthus roseus* than in the control [18]. Moreover, treating cell suspension cultures with protein PB90 can induce the accumulation of ABA, followed by the increase in the gene expression of *CrTDC* and *CrSTR*, and finally, an increase in the content of TIAs [19]. Furthermore, ubiquitination-mediated protein degradation is an important process in ABA signal transduction. [20,21]. In regulating the expression of stress-responsive genes, transcription factors, post-transcriptional modification and secondary metabolism play a vital role in responding to abiotic stress [22,23].

Protein ubiquitination, as an important post-translational modification, mediates the degradation of specific proteins and is widely involved in a series of cellular processes in many species [24,25]. Ubiquitination modification is an important post-translational modification of proteins, playing a vital role in plant development and growth [26]. Ubiquitination is a complex reaction process that involves three enzymes, E1 ubiquitin-activating enzyme (E1), E2 ubiquitin-binding enzyme (E2), and E3 ubiquitin-ligase (E3) [27]. The E1 ubiquitin-activating enzyme hydrolyzes ATP to form high-energy thioester bonds at its own active sites, thereby activating ubiquitin. The activated ubiquitin is then transferred to the E2 ubiquitin binding enzyme through cross-esterification. Subsequently, the E3 ubiquitin ligase specifically connects the ubiquitin-binding enzyme to the substrate protein. Ultimately, the target protein is hydrolyzed into smaller peptide segments on the 26S proteasome [28]. Because the E3 ubiquitin ligase can specifically recognize target proteins, it plays a decisive role in the ubiquitin pathway. According to the structural domains and ways of binding to the substrate, E3 can be classified into four categories: homologous to E6-associated protein carboxyl terminus (HECT), U-box, ‘really interesting new gene’ (RING), and cullin-RING ligases (CRLs) [29]. The CRLs are the largest superfamily of E3 ubiquitin ligases, consisting of cullin skeleton protein, RING Box 1, and substrate recognition subunits [30,31]. At present, six classic cullin proteins (AtCUL1, AtCUL2, AtCUL3a, AtCUL3b, AtAPC2 and AtCUL4) have been identified from *A. thaliana* [32].

Ubiquitination plays a crucial role in regulating ABA signaling by controlling protein stability and activity, as well as the inner membrane transport pathways involved. This regulation influences key factors involved in the synthesis and signal transduction of abscisic acid, thereby impacting the plant response to ABA [33]. The biggest challenge in understanding the mechanism of ABA signaling is identifying ABA receptors. Recent studies have demonstrated the ubiquitination of ABA receptors and ABA signaling proteins, implying that ubiquitination is important to the ABA signaling mechanism. AIP2 (ABI3-interacting protein 2) is an E3 ubiquitin ligase with a RING domain, which can ubiquitinate the downstream gene *ABI3* in the ABA signaling pathway [34]. The CUL-DDB1 E3 ubiquitin-ligase utilizes DWA1 and DWA2 as substrate recognition subunits, acting as a negative regulator of the ABI5 content. *A. thaliana* mutants of Dwa1, Dwa2 and CUL4 are more sensitive to ABA, salt and drought, and the expression of responsive genes were upregulated after ABA treatment [35]. Furthermore, the CUL E3 ubiquitin ligase plays a crucial role in reducing the sensitivity of *A. thaliana* to stress environments by degrading the ABA-inducible transcription factor ATHB6 [36]. In general, CRL is probably the most characterized E3 involved in plant growth and development. However, the mechanisms underlying the ubiquitination modification and degradation of functional proteins in regulating the synthesis of TIAs in *U. rhynchophylla* has not been reported. The genome of *U. rhynchophylla* provides the ability to analyze gene families, explore key enzyme genes and elucidate molecular regulatory mechanisms [37]. In this study, we systematically identified the *CUL* gene family members, gene structure, classification, chromosomal location and *cis*-acting elements in the genome of *U. rhynchophylla*. Finally, the gene expression of *CUL* genes and key enzyme genes were analyzed after ABA treatment with a co-expression analysis, providing a basis for the subsequent functional research of *CUL* genes.

## 2. Results

### 2.1. Identification and Characterization of CUL Gene Family

Twelve *UrCUL* genes were identified from the genome of *U. rhynchophylla*, and they were named according to their classification with AtCUL proteins (Figure 2). The physicochemical properties and subcellular localization of all the UrCUL proteins were analyzed (Appendix A). The molecular weight of the UrCULs varied from 20.17 kDa (UrCUL2-likeB) to 158.90 kDa (UrCUL1-likeB), corresponding with the number of amino acids that varied from 173 to 1367. The pI values ranged from 4.76 (UrAPC2) to 9.18 (UrCUL2-likeB), with the prediction that eight UrCUL proteins were acidic and the remaining four proteins were basic proteins (UrCUL3A, UrCUL4-likeB, UrCUL2-likeA and UrCUL2-likeB). Two-thirds of the proteins were predicted to be unstable. From the protein subcellular localization, it was predicted that most of the CUL proteins were located in the nucleus, while UrCUL3A, UrCUL4-likeA and UrCUL4-likeB were predicted to exist in the cytoplasm (Appendix A).

### 2.2. Phylogenetic Relationships of CUL Genes

To investigate the evolutionary history of the CUL genes in *U. rhynchophylla*, we constructed a phylogenetic tree using the MEGA 11.0 tool based on the CUL proteins from *U. rhynchophylla* (12 members), and *A. thaliana* (6 members) (Figure 1). According to the domains related to the function of the CUL proteins, these specific proteins linked to CUL were classified as CUL-SCF (S-phase kinase-associated PROTEIN 1 (SKP1) -CUL-F-box), CUL-BTB (Bric a BRAC, Tramtrack and Broad Complex), CUL-DDB1 (UV-damaged DNA Binding Protein 1) and CUL-APC/C (Anaphase Promoting Complex), respectively, which were divided into four groups: CUL-SCF (Group I), CUL-BTB (Group II), CUL-DDB1 (Group III) and CUL-APC/C (Group IV). In order to evaluate the degree of gene expansion or loss during evolution, the CUL genes in each group were counted. In *U. rhynchophylla*, it was found that Groups I–IV contained seven, two, two, and one CUL genes, respectively. In *A. thaliana*, Groups I to IV contained two, two, one, and one CUL genes, respectively. The increased gene number in group I and III implies the presence of gene expansion in *U. rhynchophylla.*

### 2.3. Gene Structure and Motif Analysis of CUL Genes

In order to investigate the diversity of the gene structures of *UrCUL* genes, the complete cDNA sequences of 12 *UrCUL* genes with their respective genomic DNA sequences were extracted. The sequences revealed that 12 *UrCUL* genes exhibited a variable number of exons, that ranged from 2 to 37. *UrCUL1-likeB* was found to have the highest number of exons, while the *UrCUL3A* only had two exons and one intron (Figure 3a). The variations in the number of exons among the *UrCUL* gene family might suggest diverse roles in secondary metabolite biosynthesis, as well as in the growth and development processes in *U. rhynchophylla*. The conserved motifs of the 12 *UrCUL* genes were analyzed using the MEME website. The results showed the presence of motifs 2–7 distributed among the CUL members (Figure 3b), and almost all the UrCUL proteins (81.8%, 9/11) had motifs 1-5; however, CUL2-like B only had motif 2 and motif 5. The shared motifs found within the gene sequences imply conserved evolutionary relationships and potentially similar cellular functions, providing some clues for predicting their biological function.

### 2.4. Cis-Acting Elements in the Promoters of CUL Genes

*Cis*-regulatory elements (CREs) are non-coding DNA sequences in gene promoter regions that could bind transcription factors, are critical to gene expression and widely involved in the regulation of a variety of processes [38]. The PlantCare database was introduced to predict the cis-acting elements in the putative promoter region of 2000 bp upstream promoter sequences of *UrCULs* to further explore the function of the *UrCULs*. According to the predicted data, differently distributed CREs could be divided into three broad categories; 30 kinds of cis-acting elements related to plant growth and development (135), which were phytohormone (89) and abiotic and biotic stress responsive (62) were identified (Figure 4). The first category relates to plant growth and development, including light-responsive elements (Box4 (26); G-box (17); GT1-motif (26); GATA-motif (6); ATCT-motif (3); I-box (5); AT1-motif (2) et al.; meristem-associated (CAT-box (9); O_2_-site (5); circadian and endosperm-expressing GCN4 motif (4). Especially, plant-hormone-related cis-acting elements including the abscisic acid (ABA) response elements ABREs (16); the gibberellin (GA, 9) response elements P-box (4), and TATC box (1); the salicylic acid (SA) response elements TCA (9); the methyl jasmonate (MeJA) response elements CGTCA motif (17) and TGACG motif (17); and the auxin response elements AuxRR-Core (2), CGTCA (17) and TGA-element (6) were widely present in the promoter region of most of the *UrCULs*. The cis-acting element antioxidant response such as anaerobic induction ARE (37), drought MBS (5), wound induction WUN-motif (1), low temperature LTR (11), and defense TC-rich repeats (6) were also found. The results indicate that the composition and number of *cis*-regulatory elements in the promoter regions of different *UrCULs* were diverse, associated with the transcriptional regulatory networks of abiotic and biotic stress responses, light and phytohormone responses, as well plant growth and development processes in *U. rhynchophylla*.

The greater the quantity, the darker the color. Light blue, yellow, and green represent three categories of cis-acting elements related to plant growth and development, phytohormone response, and abiotic and biotic stress.

### 2.5. Chromosomal Distribution and Collinearity Analysis of UrCUL Gene Family

The 12 *UrCUL* genes were unevenly distributed on the eight chromosomes of *U. rhynchophylla* (Figure 5b) (chr3, chr7, chr8, chr10, chr17, chr18, chr19, chr22), and the number of *UrCUL* genes on chromosome 22 was the largest, with four *UrCULs* (Figure 5a). Large fragment duplications and tandem duplication events are thought to be the main cause of gene family amplification in the genome [39]. To investigate the evolutionary relationships of the *UrCUL* family, we performed intraspecific and interspecific collinearity analyses of the *UrCUL*s from three different species (*A. thaliana, Oryza sativa, Coffea canephora*). The intraspecific collinearity analysis indicated that it has four duplicated fragments (Figure 5c). *UrCUL1-likeC*/*UrCUL1-likeD* and *UrCUL2-likeA*/*UrCUL2-likeB* were identified to be the tandem duplicated genes; all four genes are located at the same location on chr22, and studies have shown that tandem gene duplication is an important cause of gene clustering [40]. This suggested that gene duplication, particularly segmental duplication, may be responsible for the expansion of the *UrCUL* gene family, which might be the major driving force of *UrCUL* gene evolution. The interspecific collinearity analysis showed that the *UrCULs* formed ten collinear gene pairs with *A. thaliana*, eight collinear gene pairs with *C. canephora* and only one collinear gene pair with *O. sativa* (Figure 5c)*. UrCUL1* formed collinear gene pairs with the *CUL* genes of *A. thaliana*, *C. canephora* and *O. sativa*, suggesting that *UrCUL1* potentially contributed to the expansion of the *UrCUL* family.

### 2.6. Expression Analysis of UrCUL Genes under ABA Stress Treatment

In order to investigate the expression patterns of the *UrCUL* genes that may be related to the ABA stress response, we used qRT-PCR to explore the changes in the expression levels of 12 *UrCULs* during different times with 100 μM of ABA (0, 0.5, 1.0, 4.0 and 8 h). According to the expression pattern, the expression levels of 12 *UrCULs* can be roughly divided into three categories (Figure 6). The expression levels of *UrCUL1*, *UrCUL*2*-LikeA* and *UrCUL*2*-LikeB* were the highest at 0 h, and decreased first and then increased with the increase in the ABA treatment time. The expression levels of most of the *UrCUL*s did not show significant changes, while *UrCUL1-LikeB, UrCUL1-LikeC, UrCUL1-LikeD, UrCUL3A, UrCUL4-LikeA* and *UrCUL4-LikeB* showed the highest expression levels after 8 h of ABA stress treatment. The expression levels of three genes, *UrCUL1-LikeA* and *UrCUL3B*, and *UrAP2*, first increased and then decreased, reaching their peak at 1 h.

### 2.7. Expression Analysis of Key Enzyme Genes under ABA Stress

Fifteen key enzyme genes involved in the TIA biosynthesis pathway showed different expression patterns under the ABA treatment. Almost all the key enzyme genes showed a trend towards downregulation after 1 h of ABA hormone treatment, and then upregulated to the initial expression level after 8 h, including *UrGES*, *UrG8H*, *Ur7DLH*, *UrLAMT*, *UrAS*, *UrTDC* and *UrSTR*. In addition, there were six genes that were downregulated after 8 h of ABA treatment. Interestingly, the gene expression level of *UrSLS* reached its peak at 1 h of treatment, which increased four-fold compared to 0 h and then gradually returned to the original level at 8 h (Figure 7).

### 2.8. Expression Patterns of UrCUL Genes and Key Enzyme Genes under ABA Treatment

The co-expression correlation analysis between the 12 *UrCUL*s and 15 key enzyme genes in the TIA biosynthesis pathway were constructed. *UrCUL1* and *UrCUL2-likeB* were positively correlated with the variation trend of six genes (*GES*, *G8H*, *LAMT*, *AS*, *STR* and *SGD*) in the TIA pathway. At the same time, *UrCUL1-likeA* was significantly positively correlated with the expression levels of six genes (*8HGO*, *IO*, *7-DLGT*, *AnPRT*, *TSA*, *SGD*). The expression trend of *UrCUL2-likeA* and five genes in the pathway (*GES*, *G8H*, *AS*, *TSB*, *ST*R) had a highly significant positive correlation; among them, *UrCUL1-likeB* and *UrCUL1-likeC* showed a significant positive correlation (*TSB*, *TDC*) with two genes in the pathway, respectively, and a significant negative correlation with the expression pattern of *7-DLGT* and *AnPRT* (Figure 8c). Similar results have been reported in *Xanthoceras sorbifolium,* as *XsbHLH59*, *XsbHLH71* and *XsbHLH102* were also significantly upregulated under ABA stress [41].

## 3. Discussion

Cullin protein is a molecule widely found in plants that plays a crucial role in the post-translational modification of cellular proteins involving ubiquitin [42]. At present, the research on the CUL ubiquitin ligase genes was mainly based on the genetic and biochemical studies of model plants such as *A. thaliana* [43,44], and there have been preliminary studies on the secondary metabolism of *S. miltiorrhiza* in medicinal plants [37], but little is known about other plants. *U. rhynchophylla* is a traditional Chinese medicinal plant that is rich in TIAs, which have therapeutic effects on hypertension and Alzheimer’s disease [27,28,29]. However, the characteristics and functions of the CUL genes of *U. rhynchophylla* remain unclear.

In this study, a total of 12 *UrCUL* genes were identified from the *U. rhynchophylla* genome, all of which had complete N-terminal and C-terminal sequences and could be translated normally. The number of *UrCULs* was close to that of the *AtCULs* (11) and *OsCULs* (13) [42], but more than that of the *SmCULINs* (8) [28], and only six *AtCUL*s in *A. thaliana* had a complete C-terminal and N-terminal. Building a phylogenetic tree with *A. thaliana*, the CUL proteins were classified into four categories, consistent with the classification of *S. miltiorrhiza.* Specifically, CUL-SCF (Group I), Cul-BTB (Group II), Cul-DDB1 (Group III) and Cul-APC/C (Group IV) [28].

The number and length of introns and exons can be used to investigate whether gene deletion or insertion occurred during the evolution of gene families [44]. All the *UrCUL* genes contained exons, introns and CUL domains. The number of exons ranged from 2 to 37, far more than that of *S. miltiorrhiza* (1 to 19), and some *SmCUL* genes had no introns. These results suggested that the insertion of additional introns in the *UrCUL* gene family may have occurred during the evolutionary process. It has been shown that a large number of introns in the *CUL* gene have the effect of protecting the coding sequence from mutations with functional defects [45]. The exon number of each *UrCUL* gene was different, and there are five motifs in the *UrCULs* compared to three motif elements in *S. miltiorrhiza*, indicating the complexity and universality of the gene function of the *UrCUL* gene family. An interspecific and intraspecific collinearity analysis showed that tandem duplication and segmental duplication occurred in the *UrCUL* gene family, and *UrCUL1* played a crucial role in the expansion of the gene family. In addition, these *UrCUL* genes that form tandem repeat events belong to the same subfamily. Notably, in addition to gene replication, the amplification of *UrCUL* genes can also be caused by the gain or loss of introns, which may be another important source of gene amplification [46]. The differential expression of Cullins was associated with multiple stress and signal response elements in the promoter region [47]. Thirty different cis-acting elements were predicted for the promoter region of the *UrCUL*s family, among which 135 motifs were related to plant growth and development, and included plant-hormone-responsive motifs (89), and abiotic and biological motifs (62) (Figure 4). However, there are only 16 types of *cis*-acting elements in rice and 18 types in *S. miltiorrhiza*. These findings suggest that *UrCLUs* may be regulated by various cis-acting elements in the promoter during the growth and stress response, and may play more complex and diversified functions. As a key cis-acting element in response to ABA treatment, it has been identified that 9 out of the 12 *UrCULs* promoter regions contained ABA-responsive elements (including *UrCUL1*, *UrCUL1-likeA*, *UrCUL1-likeB*, *UrCUL1-likeC*, *UrCUL1-likeD*, *UrCUL2-likeA*, *UrCUL4-likeA UrCUL4-likeB*, *UrAPC2*), which indicates that UrCULs may play an important role in ABA signal transduction in *U. rhynchophylla*. In addition, the GT1 motif and ARE exist in all the *UrCUL* genes, indicating that all the *UrCULs* are closely related to the light response and antioxidant processes in *U. rhynchophylla*. Interestingly, the numbers of CGTCA motifs and TGACG motifs in all the *UrCULs* promoters were completely consistent, indicating that they may have a sensitive effect on the plant hormone response of *U. rhynchophylla.*

The plant hormone ABA plays a role in multiple pathways such as stress response, plant development, and secondary metabolism [48]. In *A. thaliana*, *AtCUL3* interacted with *AtHB6* in response to ABA induction [49]. When the hairy roots of *S. miltiorrhiza* were treated with ABA, this resulted in the expression of *SmCUL* genes significantly changing, especially the *SmAPC2*, *SmCUL2* and *SmCUL4* genes, which may be involved in the biosynthesis of phenolic acids and tanshinones [28]. In *Actinidia chinensis*, the transgenic ectopic expression of *AcSINA2* in *A. thaliana* resulted in the seedlings exhibiting a hypersensitive growth phenotype to ABA treatment [50]. In addition, ABA also might influence the gene expression of the TIA biosynthesis pathway and subsequently stimulate the content of indole alkaloids in *C. roseus* and taxol in *Taxus chinensis* [51]. Therefore, it appears that ABA as a signal might involve an ABA increase and the activation of transcription factors containing ABRE motifs, followed the key enzyme gene expression increases and ultimately causing accumulation in secondary metabolites. On the other hand, there is increasing evidence that the ubiquitination system plays a regulatory and maintenance role in plant transcriptional regulation [52,53], and there are studies targeting genes involved in the protein ubiquitination system, especially the components of E3 ligase [54], in order to develop stress-tolerant transgenic plants [55,56]. Since ABA response *cis*-elements occurred most frequently in the promoter of the *UrCULs* (Figure 4), this prompted us to investigate the gene expression pattern of *UrCULs* in response to ABA treatment.

The analysis of gene expression patterns is one of the most important means of studying gene function [48]. Therefore, the response of the 12 *UrCUL* genes to the ABA hormone was studied, and it was found that eight *UrCUL* genes containing ABRE *cis*-acting elements in the promoter region had significant changes in response to the ABA hormone compared with those exposed for 0 h (Figure 5). The *UrCUL* genes without ABRE cis-acting elements in the *UrCUL3A* and *UrCUL3B* promoter regions had no response to the ABA treatment. A transcript profiling analysis in rice showed that 13 *OsCUL* genes were involved in the growth and development of rice plants, and all genes responded to ABA stress [43]. The results showed that all had altered expression throughout the experiment, and that six *UrCUL* genes were maximally expressed under ABA stress after 4 h. In addition, several *UrCUL* genes were highly expressed at 4 h, returning to the initial expression level after 8 h of cold treatment, indicating that these *UrCUL* transcription factors may have diverse functions at different periods of the stress response. Under ABA treatment, plants respond to ABA hormones, and the CULLIN-E3 ubiquitin ligase may act on key enzyme genes in the TIA pathway of U. rhynchophylla through protein ubiquitin degradation, causing most genes in the pathway to decrease within 1–2 h. As the processing time increases, the degree of ubiquitination gradually decreases, causing the expression level of key enzyme genes in the pathway to gradually increase to the initial level. For example, *UrG8H*, *UrG8H*, *UrSLS*, *UrTSB*, *UrTDC* and *UrSTR* were significantly upregulated under ABA stress, while the expression of *UrAS* and *UrTDC* were increased but not significantly, compared to cytokinin (CK). Interestingly, *Ur10HGO*, *UrIO*, *Ur7DLGT*, *UrAnPRT*, *UrTSA* and *UrSGD* were highly upregulated compared to cytokinin (Figure 7b). A correlation analysis between the *UrCUL* genes and key enzyme genes was performed (Figure 7c). The results showed that *UrCUL1*, *UrCUL1-likeA*, *UrCUL2-likeA and UrCUL2-likeB* were positively correlated with key enzyme genes. Among these, *UrGES*, *UrG8H*, *UrAS*, *UrSTR* and *UrSGD* showed strong correlations (*p* < 0.05, *r* > 0.8) with *UrCUL1*, *UrCUL1-likeA* and *UrCUL2-likeB*. These results indicated that *CUL* genes and key enzyme genes could share similar regulation by the ABA treatments, implying that these *UrCUL* genes might participate in TIA biosynthesis.

## 4. Materials and Methods

### 4.1. Plant Materials and ABA Treatments

The plant tissue was obtained from the College of Horticulture, Hunan Agricultural University, Changsha, Hunan Province, China, and identified by Prof. Shugen Wei. Seedlings of the plant were cultured in tissue culture for 180 days before use. The media composition for plant tissue culture was 1/2 MS medium, with 3-indolebutyric acid: 0.2 mg/L; 1-naphthlcetic acid: 0.2 mg/L; sucrose: 25 g/L; agar: 4.5 g/L; and activated carbon: 0.5 g/L. Greenhouse conditions were maintained at 25 °C during the day and 18 °C at night, with a 12/12 h (light/dark) light intensity of 120–150 µmol/m^2^/s and 60% humidity. After uniform growth of about 180 days, roots of *U. rhynchophylla* were treated with 100 μM concentration of ABA and samples were gathered at 6, 12, 24, 36 and 48 h with 0 h as control. Fresh roots of *U. rhynchophylla* treated at different time periods were collected and immediately stored in liquid nitrogen and then stored in a −80 °C freezer for follow-up experiments.

### 4.2. Identification of CUL Genes in U. Rhynchophylla

The *U. rhynchophylla* genome was recently sequenced by Oxford Nanopore technology, with a genome size of 627 Mb and a contigN50 of 1.8 Mb (not published). This genome sequence provides a large genetic resource for the analysis of the *U. rhynchophylla* TIA pathways. The CUL domain (PF00888) retrieved from Pfam (http://pfam.xfam.org, accessed on 11 November 2023) [57] was used to identify the potential *CUL* genes of *U. rhynchophylla*. The NCBI Conserved Domain Database (CDD) (https://www.ncbi.nlm.nih.gov/Structure/bwrpsb/bwrpsb.cgi, accessed on 11 November 2023) and the online website Expasy Prosite (https://prosite.expasy.org/, accessed on 11 November 2023) were used to further verify the domain of UrCUL. In addition, Expasy Protparam (https://web.expasy.org/protparam/, accessed on 11 November 2023) was introduced to calculate the physical and chemical properties. Subcellular location analysis of CUL candidate genes were completed by Cell-PLoc 2.0 (http://www.csbio.sjtu.edu.cn/bioinf/plant-multi/, accessed on 13 November 2023) [58].

### 4.3. Multiple Sequence Alignment and Construction of Phylogenetic Tree

The *U. rhynchophylla* genome database obtained by genome sequencing. The *CUL* protein sequences from *A. thaliana* were acquired from the TAIR database (https://www.arabidopsis.org/index.jsp, accessed on 11 November 2023). Multiple sequence alignment was performed through MAFFT software. The sequences of CUL protein from *U. rhynchophylla* and *A. thaliana* were compared, and a phylogenetic tree was constructed using MEGA 7.0 software with neighbor-joining method [59].

### 4.4. Conserved Motif, Gene Structure and Cis-Regulatory Element Analysis

To visualize the *UrCUL* gene structures (introns and exons), TBtools v2.109 software was utilized with *U. rhynchophylla* genome database. Conserved motifs of *CUL* gene were predicted by using online software MEME (https://meme-suite.org/meme/, accessed on 17 November 2023), with the following conditions: maximum number of motifs = 5; the length of motifs ranging from 20 to 50. The upstream 2000 bp DNA sequences of the identified *UrCULs* were extracted by TBtools, which were submitted to the PlantCARE (http://bioinformatics.psb.ugent.be/webtools/plantcare/html/, accessed on 22 November 2023) database to extract *cis*-regulatory elements (CREs) [60].

### 4.5. Chromosome Localization, Syntenic Analysis and Gene Duplication Events

In order to obtain the location of *UrCUL* genes on the chromosome, information about the physical location of the gene was collected from the genome of *U. rhynchophylla*, it was mapped to the 10 chromosomes by TBtools (v2.109), and named according to their classification with AtCUL proteins. Gene duplication events were employed by Multiple Collinearity Scan toolkit (MCScanX) in TBtools with the following parameters: CPU for BlastP: 8; E-value: 1 × 10^−10^ [61]. Collinearity analysis between *U. rhynchophylla* and two other species (*A. thaliana*, *O. sativa*) were performed using TBtools.

### 4.6. Total RNA Extraction, cDNA Synthesis and qRT-PCR Validation

The total RNAs from *U. rhynchophylla* leaves were isolated according to the manufacturer’s instructions via the SteadyPure Plant RNA Extraction Kit (Accurate Biology, Hunan, China). The primers used for qRT-PCR Specific primer pairs designed using Beacon Designer 7.0 software are listed in Appendix A; the primers of key enzyme genes involved in TIA biosynthesis pathway have been reported previously with the *SAM* gene as the internal reference gene [62]. The qRT-PCR was performed using 2× SYBR Green Pro Taq HS premix (Vazyme, Shanghai, China) and reaction system and procedure have been previously reported [63], using the following protocol: 95 °C for 3.00 min, 1 cycle; 95 °C for 15 s and 60 °C for 30 s, 40 cycles. The 2^−ΔΔCt^ method with three replications was performed for analysis and the statistical tests were performed using GraphPad Prism 9.0 software. Correlation expression analysis between *UrCULs* and key enzyme genes was performed using the OmicStudio online website (https://www.omicstudio.cn/, accessed on 25 November 2023), significant difference was determined based on Student’s *t*-test (*p* < 0.05 and *p* < 0.01) and marked with one and two asterisks, respectively.

## 5. Conclusions

In the current study, a genome-wide survey and evolutionary analysis of the *UrCUL* gene family in *U. rhynchophylla* have been conducted for the first time. The expression analysis of the *UrCUL* genes was measured after ABA treatment, with some responding, implying they may function in response to ABA stress. The results of a co-expression network showed that six *UrCUL* genes might participate in TIA biosynthesis, which provides a theoretical basis for understanding the biological functions of the *CUL* gene in plant secondary metabolites, growth and development.

## Figures and Tables

**Figure 1 plants-13-00532-f001:**
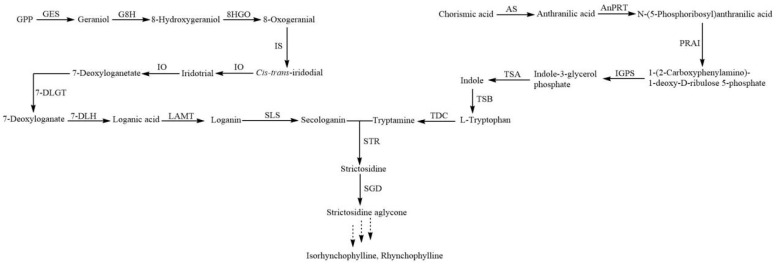
Terpenoid indole alkaloid biosynthesis pathway. The solid arrow signifies that the biosynthesis pathway is confirmed. The dashed arrow signifies that the biosynthesis pathway is unknown.

**Figure 2 plants-13-00532-f002:**
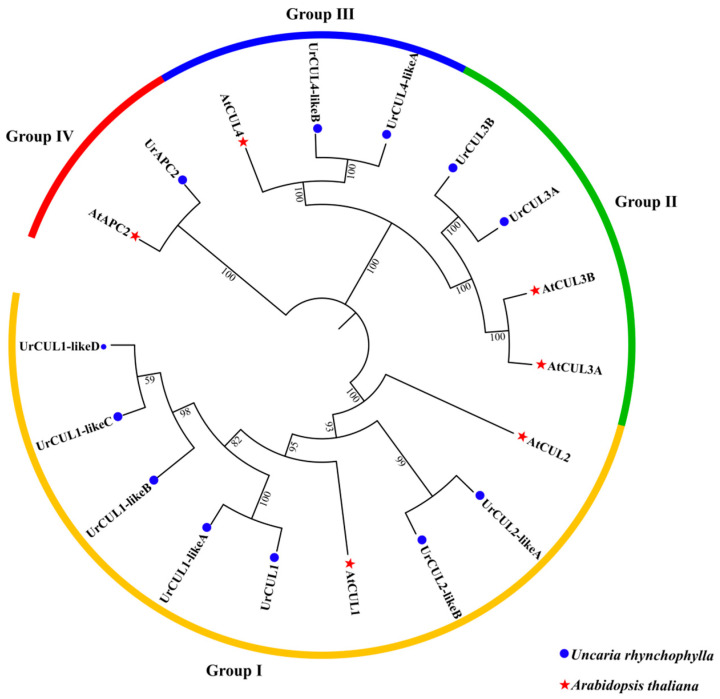
Phylogenetic tree of CUL proteins from *A. thaliana* and *U. rhynchophylla* was aligned by ClustW, and the phylogenetic tree was constructed using the neighbor-joining method and MEGA 7.0 with 1000 bootstrap replicates.

**Figure 3 plants-13-00532-f003:**
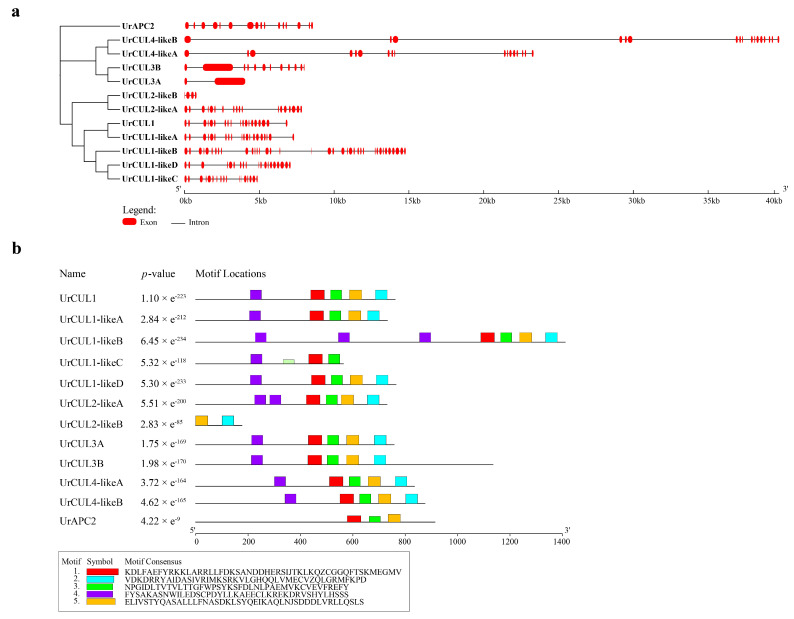
The conserved motifs and gene structure analysis of CUL genes in *U. rhynchophylla*. (**a**) The exon/intron distribution of the twelve *U. rhynchophylla* CUL ubiquitin ligase genes were determined using the GSDS tool by comparing the coding sequences (CDSs) with the relative genomic sequences. The red box represents the CDS; the solid black line depicts the intron region. (**b**) Schematic representation of five motifs discovered in *U. rhynchophylla* CUL ubiquitin ligase genes through MEME website denoted by different colors.

**Figure 4 plants-13-00532-f004:**
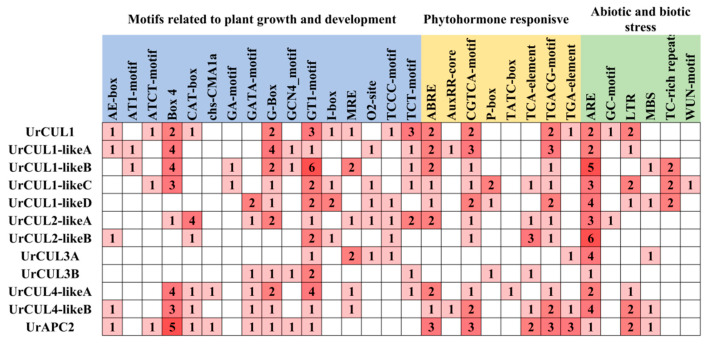
The *cis*-acting element analysis of putative promoter of twelve CUL genes.

**Figure 5 plants-13-00532-f005:**
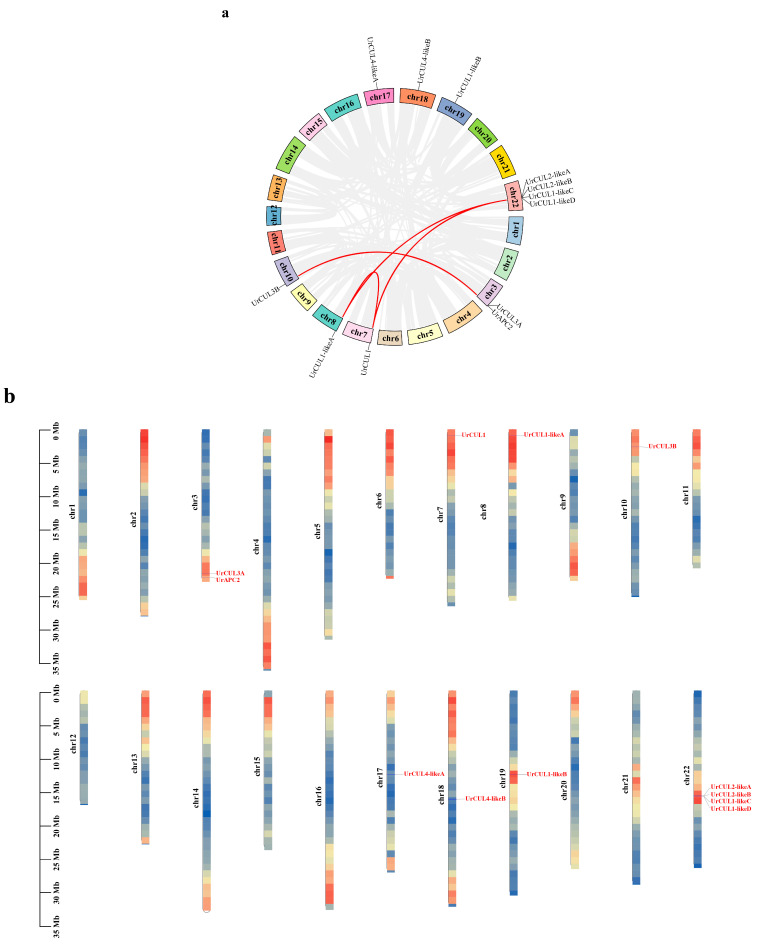
The chromosomal location and collinearity relationship of *UrCULs*. (**a**) *UrCULs* are marked on chromosomes. The scale bar on the left indicates the length of *U. rhynchophylla* chromosomes (Mb). (**b**) Syntenic analysis of *UrCULs*. The circle plot was created with the MCScanX tool. Identified collinear genes are linked by brown lines. (**c**) Syntenic relationship of *CULs* among *U. rhynchophylla*, *A. thaliana*, *O. sativa*, *C. canephora.* Identified collinear *UrCULs* are connected by red streaks.

**Figure 6 plants-13-00532-f006:**
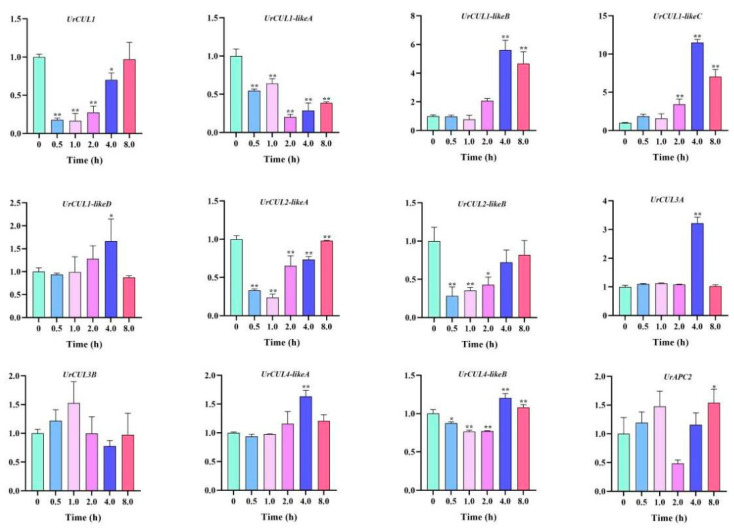
Expression analysis of 12 selected *UrCUL* genes in *U. rhynchophylla* leaves under ABA treatment. The final results are expressed as mean ± standard deviation. * *p* < 0.05, ** *p* < 0.01.

**Figure 7 plants-13-00532-f007:**
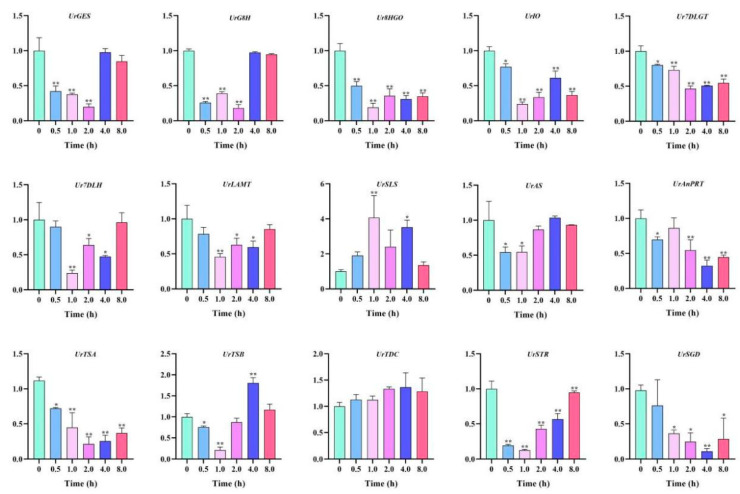
Expression patterns of the key enzyme genes under ABA treatment. The final results are expressed as mean ± standard deviation. * *p* < 0.05, ** *p* < 0.01.

**Figure 8 plants-13-00532-f008:**
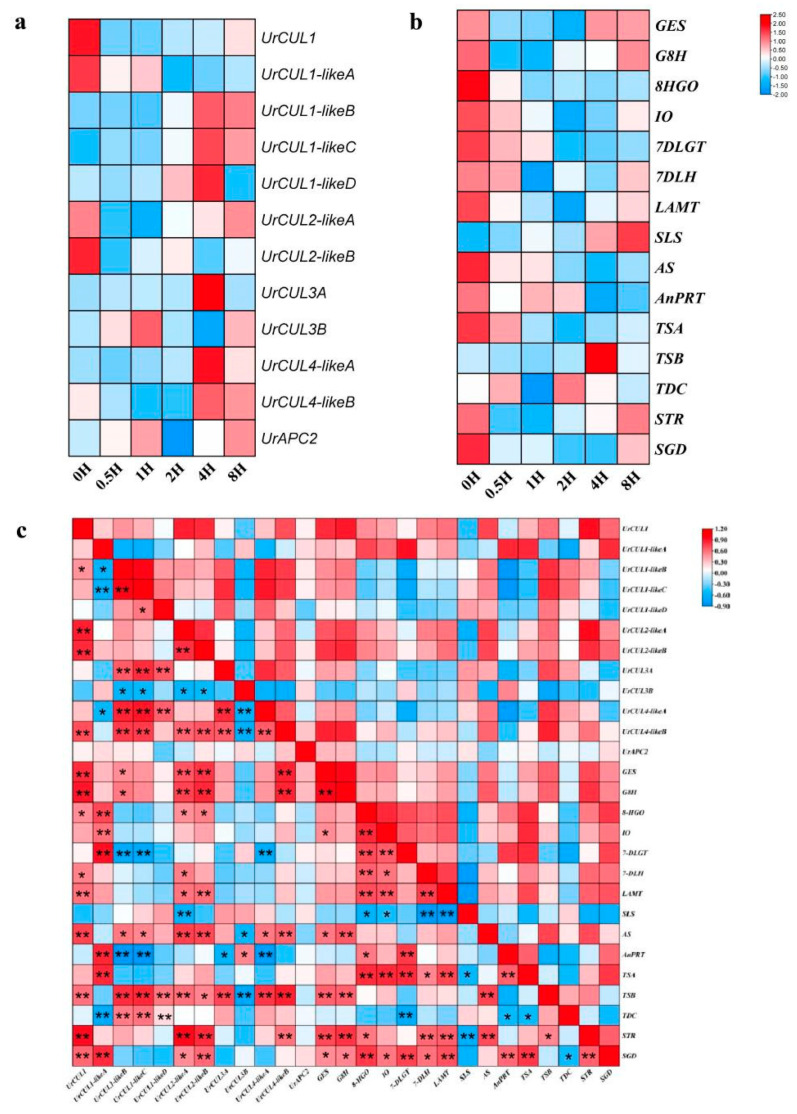
Expression patterns of the *UrCUL* genes and key enzyme genes under ABA treatment. The heatmaps show the expression level ratio at 0 h, 0.5 h, 1 h, 2 h, 4 h and 8 h. The color represents expression levels from upregulation (red) to downregulation (blue). (**a**): The expression pattern of *UrCUL* genes in the roots of *U. rhynchophylla*; (**b**): the expression pattern of the key enzyme genes in the roots of *U. rhynchophylla*; (**c**): the correlation between the gene expression profiles of *UrCUL* and key enzyme genes under ABA treatment. The red represents the positively correlated, the blue presents the negatively correlated. * represents *p* < 0.05 and ** represents *p* < 0.01.

## Data Availability

All relevant data are within the manuscript and its Appendix A.

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
