# Peer review of "Genome-Wide Mining of CULLIN E3 Ubiquitin Ligase Genes from Uncaria rhynchophylla"

_plants, 2024, doi:10.3390/plants13040532_

Round 1
Reviewer 1 Report
Comments and Suggestions for Authors
An intriguing report on a plant which contains medically-useful alkaloids. Two main suggestions--regarding your introduction and regarding your affirmations about the TIA pathways. Introduction--on the attached manuscript, I've made several suggestions (and asked some clarifying questions). Your introduction, in my opinion, is strong at present--but could be stronger if you can address my reviewer questions. Secondly, TIA pathways--I have asked the same question at several points if the known enzymatic steps in the pair of pathways you present (I like the figure that you use, by the way). My question is this--have those steps been shown in your target plant? Or have they been documented in other plant species? Either way, I believe that your work is sound--but I do believe that you should clarify this point.

Comments on the Quality of English LanguageEnglish usage is, in this reviewer's opinion, sound. I have marked a very small number of places where revision could be helpful (on the attached document).
Reviewer 2 Report
Comments and Suggestions for Authors
Revision
In this article, authors identified 12 UrCUL (Cullin subtype of E3 ubiquitin ligase from U. rhynchophylla) gene members from U. rhynchophylla genome identified and classified into 4 subfamilies based on a phylogenetic relationship with CULS (a subtype of E3 ubiquitin ligase) from A. thaliana. After identification, authors treated roots of U. rhynchophylla tissue culture seedlings with abscisic acid (ABA) and analyzed the expression pattern of UrCUL genes. They found that UrCULs gene expression was ABA-dependent and that they could be involved in response to ABA hormones, therefore having an important role in plant reaction to stress, development, and secondary metabolism, especially 6 of them on terpenoid indole alkaloid synthesis (TIA). Since these results could be of importance for understanding mechanism of TIA biosynthesis and help us umprove their production in the U. rhynchophylla plant this article could be published after some minor revisions.
Minor revisions:
1. Figure 6. And 7. should have a higher resolution so that the legend can be more readable.
2. The authors could more specifically explain how these findings can help us in understanding the mechanism of TIA synthesis or help us improve their production as secondary metabolites.
